# Equine Influenza Virus and Vaccines

**DOI:** 10.3390/v13081657

**Published:** 2021-08-20

**Authors:** Fatai S. Oladunni, Saheed Oluwasina Oseni, Luis Martinez-Sobrido, Thomas M. Chambers

**Affiliations:** 1Texas Biomedical Research Institute, San Antonio, TX 78245, USA; kanmi01@gmail.com (F.S.O.); LMartinez@txbiomed.org (L.M.-S.); 2Department of Veterinary Microbiology, University of Ilorin, Ilorin P.M.B. 1515, Nigeria; 3Department of Biological Sciences, Florida Atlantic University, Davie, FL 33431, USA; soseni2013@my.fau.edu; 4Department of Veterinary Science, Gluck Equine Research Center, University of Kentucky, Lexington, KY 40546, USA

**Keywords:** equine influenza, equine influenza virus, equine influenza vaccine, H3N8, adaptive immunity, cellular immunity, humoral immunity, surveillance, experimental infection

## Abstract

Equine influenza virus (EIV) is a constantly evolving viral pathogen that is responsible for yearly outbreaks of respiratory disease in horses termed equine influenza (EI). There is currently no evidence of circulation of the original H7N7 strain of EIV worldwide; however, the EIV H3N8 strain, which was first isolated in the early 1960s, remains a major threat to most of the world’s horse populations. It can also infect dogs. The ability of EIV to constantly accumulate mutations in its antibody-binding sites enables it to evade host protective immunity, making it a successful viral pathogen. Clinical and virological protection against EIV is achieved by stimulation of strong cellular and humoral immunity in vaccinated horses. However, despite EI vaccine updates over the years, EIV remains relevant, because the protective effects of vaccines decay and permit subclinical infections that facilitate transmission into susceptible populations. In this review, we describe how the evolution of EIV drives repeated EI outbreaks even in horse populations with supposedly high vaccination coverage. Next, we discuss the approaches employed to develop efficacious EI vaccines for commercial use and the existing system for recommendations on updating vaccines based on available clinical and virological data to improve protective immunity in vaccinated horse populations. Understanding how EIV biology can be better harnessed to improve EI vaccines is central to controlling EI.

## 1. Recent Outbreaks of EIV

The transboundary transport of horses has allowed infectious disease pathogens to cross geographical barriers between nations of the world [1,2], enabling their spread to disease-free countries (Figure 1). During horse events such as horse races and shows, horses from different parts of the world congregate and are sometimes held at high stocking density, enabling close contacts and predisposal to EIV. The importation of subclinically infected carrier horses and poor biosecurity/quarantine measures can facilitate the spread of EIV to naïve horse populations, leading to major outbreaks.

**1986–1999:** EIV was long recognized as an important equine respiratory pathogen in Europe and North America, but three outbreaks near the end of the 1980s brought it into new prominence. First, a 1986 outbreak in South Africa [3], where H3N8 EIV had previously been unknown and occurring at just the time that nucleotide sequencing was becoming commonplace, triggered the first important genetic evolutionary studies of the EIV H3 hemagglutinin (HA) [4,5] and implicated newly imported horses. In 1989 occurred the large European outbreak associated with Suffolk/89 virus [6], as well as a large-scale outbreak in China of H3N8 avian influenza in horses [7] that, fortunately, did not persist. An outbreak in Hong Kong in 1992 [8] was intensively investigated and provided the first indication of the bifurcation of the American and Eurasian lineages.

**2000–2010:** South Africa, which had eradicated EIV by 1987, suffered its second EIV outbreak in 2003, again from imported horses, which yielded the prototype Florida clade 1 (FC-1) strain, influenza A/equine/South Africa/2003 [9]. Australia, which had historically been free from EIV, endured a huge outbreak of FC-1 EIV in 2007, affecting 70,000 horses [10]. The source was imported horses that had been vaccinated but were still subclinically infected, and the virus somehow escaped from the postimport quarantine facility into the domestic population, which had never been vaccinated. This outbreak was eradicated within six months, and Australia returned to EIV-free status due to a combination of stringent horse (movement restrictions and a policy of targeted and ring) vaccination using exclusively vaccines that allow distinguishing between infected and vaccinated animals (differentiation of infected from vaccinated animals (DIVA)-capable vaccines) [11].

**2010–2021:** Over the last decade, multiple outbreaks of EIV infection were reported in many countries from different continents around the world (Table 1 and Figure 2). Increased outbreaks of EIV were reported not only in North America [12], especially in the United States (US) where the disease is endemic, but also in Europe [13,14,15], Africa [16,17,18], Asia [12,19,20], and South America [21,22,23,24,25,26]. Recently, in the US, outbreaks were reported in 23 states in 2015, 16 states in 2016, 22 states in 2017, and 33 states between the end of 2018 and the first third of 2019 [12,27]. EIV is not a notifiable disease in the US, so under-reporting is likely. In Europe, outbreaks have been reported over the years in France, Germany, Ireland, Sweden, and the United Kingdom (UK) [12]. The most recent outbreak of EIV across Europe was quite extensive, affecting 228 horse premises in the UK, 80 in Ireland, and 60 in France from the end of 2018 to 2019 according to the World Organization for Animal Health (OIE) report [27]. These outbreaks occurred among both vaccinated and unvaccinated horses, with reduced clinical manifestation observed in vaccinated horses, especially those with a history of appropriate and up-to-date vaccination over several years. This highlights the importance of maintaining homogeneity between vaccine strains and the circulating EIV strains to induce a long-lasting immunity in vaccinated horses. The observation of disease in vaccinated horses may also indicate a possible vaccine breakdown necessitating vaccine verification and appropriate use or administration to forestall future outbreaks. Outbreaks of EIV are uncommonly reported in Africa, but recently, beginning from November of 2018 to 2019, outbreaks were reported in many African countries affecting horses and donkeys [16,17,27]. This unusual epizootic outbreak was responsible for the mortality of over 66,000 horses and donkeys across Burkina Faso, Chad, Cameroon, The Gambia, Ghana, Mali, Niger, Nigeria, and Senegal [17]. EIV outbreaks have also been recently reported in Asian and Middle Eastern countries [12]. Multiple cases of EIV have been described in Japan and Mongolia from horses imported during 2007–2010 from Canada and Belgium during quarantine [28]. In 2012, 3 out of the 18 endurance horses, with up-to-date vaccine history, imported to Dubai from Uruguay displayed clinical symptoms of EIV [22]. These horses were further confirmed positive for EIV. These cases underscore the importance of international/transboundary transport of horses as the primary route of spread of EIV, even in vaccinated individuals, to EIV-free countries. Recent widespread outbreaks of EIV either in horses or donkeys have also been well documented in China [27,29], Kazakhstan [30], and Malaysia [12]. Although occasional outbreaks of EIV have been reported in South America in the past, there are pointers to a recent surge in the number of cases over the past few years [22].

Data from the genetic characterization of the HA and neuraminidase (NA) genes from EIV isolates from China, France, Ireland, Niger, Nigeria, the UK, the US, Senegal, and Sweden between 2019 and 2020 have been made available. The EIV isolates from China were characterized as the Florida clade 2 (FC-2) of the American lineage, resembling those identified in China in 2015 and 2017, while all other characterized EIV isolates belonged to FC-1, similar to the 2018 isolates from the US and South America [27].

## 2. Evolution of EIV

EI is an acute, highly contagious viral disease of equids characterized by the development of pyrexia and respiratory signs including coughing and serous nasal discharge; swollen submandibular lymph nodes may also be observed within a day or two in more severe cases [36]. EIV, the etiological agent of EI, is an enveloped, negative-sense, single-stranded, and segmented RNA virus (Figure 3). The biology of EIV, and the roles played by individual viral protein during the process of viral replication and infection, has been reviewed [37,38,39,40,41,42,43,44,45,46,47,48,49,50]. The functional relevance of each of the eight genome segments of EIV is summarized in Table 2.

Influenza A viruses (IAV) are believed to have evolved from a common ancestral progenitor, avian influenza, that diverged into continuous IAV subtypes infecting different animal species, including horses and humans [73]. The primary natural reservoirs of IAV are the aquatic birds, and both EIV and avian IAV share same host cell surface receptors. Although outbreaks of respiratory diseases in horses resembling influenza were reported in the 17th century [74] and even earlier [75], the first isolation of an EIV was that of an H7N7 configuration (subtype 1) in 1956 in Czechoslovakia, designated influenza A/Equine/1/Prague/56 [76]. This subtype was responsible for EI epizootics that spread to many parts of the world by the early 1960s. Shortly thereafter, EIV of a second subtype (subtype 2), designated as influenza A/Equine/2/Miami/63 (H3N8), was isolated from horses imported to the US from South America [77]. Both EIV subtypes cocirculated in horses across many parts of the world for over 15 years and reassorted between them during that time [78,79,80]. However, the H7N7 subtype stopped producing confirmed outbreaks of clinical disease in horses after 1979 [74,81]. This virus is now presumed to be extinct in nature. However, seroepidemiological reports, most recently from Israel [82], suggest that H7N7 EIV may still circulate; this is discussed further below under.

While the H7N7 EIV seems to have disappeared, the H3N8 EIV still causes major economic impacts to the equine industry in most parts of the world. Horses travel internationally more than any other domestic animal, and EIV is the pathogen most frequently responsible for transboundary equine disease outbreaks following horse importation [2]. Only New Zealand and Iceland have significant horse populations that have remained continuously free from EI. Some countries, e.g., Australia and South Africa, have eradicated EI after past outbreaks, but in Europe, North and South America, and Asia, EI is generally considered to be enzootic. EI vaccination is encouraged within EI-enzootic countries, although coverage is limited: e.g., within the US, EI vaccination is recommended for horses at risk, according to the American Association of Equine Practitioners (AAEP). Performance horse organizations may adopt more stringent requirements. In EI-free zones such as Australia, in which EI is a notifiable exotic disease, documented vaccination is mandatory for horses traveling there, but vaccination of resident horses is forbidden except in a state of emergency, as it conflicts with serosurveillance for the disease (see www.animalhealthaustralia.com.au/ausvetplan/, accessed 2 August 2021). DIVA-capable vaccines are the best choice in those zones, but until the 21st century, there were none available.

The evolution of EIV H3N8 subtype is driven by antigenic drift [83], the inherent ability of its HA and NA genes to mutate and accumulate point mutations over time, similar to what is commonly observed with other IAV (Figure 4). “Antigenic shift” leading to emergence of new EIV subtypes has not been observed (reassortment between the H7N7 and H3N8 EIV did not change their surface antigens). The H3 HA of EIV has reportedly drifted at the rate of only 0.8 amino acids/year [84,85,86]. This rate is less than that of human IAV, especially the H3N2 subtype. For two decades since its first report, EIV H3N8 subtype evolved as a single lineage [4]. However, EIV outbreaks of the late 1980s, whose prototype is influenza A/equine/Suffolk/1989, produced variants of circulating H3N8 leading to their divergence into an “American” and a “Eurasian” lineage (Figure 4) [38,85]. That is, geographic distance produced different virus variants in different regions, even though North America and Europe were not geographically isolated in terms of horse movements. The American lineage soon spread to Europe, while to this day, the only incursion of the Eurasian lineage into North America was a single outbreak in Canada in 1990 [87]. These variant lineages were sufficiently different that strains of both were recommended for inclusion in vaccines [88,89]. With international travel of horses for breeding and global sports events, both lineages of H3N8 were transported to other continents where EI outbreaks were linked to either the American or the Eurasian lineage.

Another bifurcation occurred around 2000, as molecular phylogenetic analysis revealed that the circulating H3N8 American lineage strains had evolved into South America, Kentucky, and Florida lineages (Figure 4) [90]. The Florida lineage further evolved into two antigenically distinct clades: Florida clade (FC)-1 and FC-2 (Figure 4) [91]. Both clades spread internationally and by 2008 had replaced the original American and Eurasian lineages. Curiously, the FC-2 clade has been absent from North America except for horses recently imported from Europe [92,93]. A variant of the FC-2 clade arose in China in 2011 and appears to have persisted [94]; it will be of interest to watch the further evolution of this variant.

Multiple amino acid substitutions in HA have occurred in the course of the evolution described above, but at the level of antigenicity, only a small number appear to be critical for emergence of successive lineages and clades. Woodward et al. [95] and Nemoto et al. [96] confirmed that changes in antigenic site B seemed to be of greatest importance although outbreaks were abetted by an additional change in a different antigenic site and that just three substitutions at positions 159, 189, and 227 were critical to the emergence of the Suffolk/89 variant that rendered the then-existing vaccines (with 1979–1981 strains) antigenically obsolete. Murcia et al. [80] suggested that increased vaccination in the 1980s resulted in an EIV evolutionary bottleneck, which was broken by the emergence of Suffolk/89 virus. Similarly, FC-1 and FC-2 originally differed by only two antigenically critical residues, at position 159, again, and position 78 in antigenic site E [91]. Recent studies [96,97] have also shown that variation at position 144, which has been occurring in FC-2 strains since 2011, reduces the cross-neutralization activity of convalescent horse sera by up to eight-fold. Murcia et al. [80] have shown by a more detailed analysis that the evolution of the EIV H3 HA since 1963 can be described as successive or cocirculation, with intrasubtypic reassortment, of 13 distinguishable clades, the most recent 2 being the FC-1 and FC-2. Further, they showed a parallel evolution of all the other viral RNA segments, with NA and PA genes evolving at rates similar to HA.

## 3. EI Vaccines

The two forms of EI vaccines currently marketed in the US are inactivated (killed) and modified live vaccines [98] (Table 3). However, other EI vaccines licensed for use in horses in other parts of the world also include recombinant virus vector vaccines and subunit vaccines [99]. DNA vaccines and other viral vectored vaccines are newer experimental vaccine technologies (Figure 5). The ability of the host to induce an EIV-specific antibody response is a correlate of protection that helps to limit the spread of the disease and viral shedding during an EI outbreak [100,101]. As well as being immunogenic, the surface glycoproteins (HA and NA) of EIV play vital roles in facilitating infection and viral spread. EIV-specific neutralizing antibodies (NAbs) directed against HA or NA represent the first adaptive defensive strategy of the host against virus infection or immunization. Induction of EIV-specific NAbs helps to neutralize virus infection by preventing virus attachment to respiratory epithelial cells (HA NAbs) and also helps in blocking virus release from infected cells (NA NAbs) [102,103]. Equally important for the control of obligate intracellular pathogens like EIV is cell-mediated immunity (CMI), which involves cytokine-mediated induction of antigen-specific cytotoxic T-lymphocytes (CTL), natural killer cells, and macrophages. Below, we evaluate each vaccine line technology with the focus on the underlining principles behind each vaccine, including its advantages and disadvantages (Table 4) and field data as a guide for appropriate vaccine selection against annual EI outbreaks.

## 4. Inactivated Whole Virus EI Vaccines

An inactivated whole virus EI vaccine consist of a whole EIV strain(s) that have been grown in mammalian culture or in the amniotic cavity of fertile hens’ eggs and then denatured either by physical (heat or irradiation) or chemical (formalin treatment) means. Virus particles in this vaccine are not able to replicate since they are destroyed, but the proteins retain antigenic epitopes recognizable by the host immune system and are able to evoke adaptive immunity. Typically, inactivated whole virus EI vaccines need to be administered multiple times for a stable protective adaptive immune response to be induced. In EIV-naïve horses, a complete primary course of vaccination requires three doses based on AAEP guidelines [98], consisting of a priming dose, a booster within 3–6 weeks, and a second booster within 5 months. Thereafter, annual boosters are recommended for most horses and twice-annual boosters for horses at risk of exposure (i.e., horses traveling for performance or breeding). Cullinane et al. [104] discuss the relevant vaccination guidelines of the OIE, Federation Equestre Internationale (FEI), and International Federation for Horseracing Authorities (IFHA) and show experimentally that, for shipment of young horses up to 4 years of age, a booster within 90 days before shipment is best, with an optimal interval of 14 days between booster and shipment. The nature of the immune response is often times similar to that of the natural infection, but this vaccine preferably instructs for the development of humoral immunity, with minimal CMI. In young horses, the serum antibody titer declines rapidly after the primary two vaccinations, and as a result, these horses may not be adequately protected for a period before the third booster dose [105,106,107]. This period between the second and third vaccinations when young horses become vulnerable is referred to as the immunity gap [106,108]. Closing the immunity gap is a concern, because this is the age in which performance horses from different stables are comingled for training, presenting elevated risk of disease exposure. Different vaccines and vaccination schedules have shown mixed success [109,110,111]. Maternal antibody interference with foal EIV vaccinations is a concern, especially with inactivated virus vaccines [110,112,113,114,115], such that the recommended age for the first (priming) dose is typically 6 months in foals whose dams are expected to be EI-seropositive. Fortunately, in such foals, maternal antibodies from colostrum are protective, and EI is seldom a problem until after 6 months of age.

One major drawback of inactivated vaccines is that they tend to be less immunogenic and, as a result, require the addition of an adjuvant to achieve protective immunity [116]. Inactivated whole virus EI vaccines are adjuvanted with the goal of enhancing or boosting the horse’s specific immune responses. An adjuvant is a chemical component of an inactivated vaccine that potentiates the immune responses to interact either physically or chemically with the antigenic component of the vaccine. Several chemical ingredients have been tried as adjuvants in inactivated EI vaccine compositions, and aluminum-based adjuvants, particularly aluminum hydroxide, continue to be widely used [117,118,119]. Aluminum hydroxide preferentially drives T_H_2-type, humoral-mediated immune responses [120,121]. Although protection against EIV is mainly by virus neutralization by vaccine-induced NAbs [122], effective elimination of EIV-infected host cells requires the protective effects of CMI. It is not clearly understood whether aluminum hydroxide adjuvanted inactivated EI vaccines can stimulate CMI. One study with an inactivated whole virus EI vaccine adjuvanted with aluminum phosphate or aluminum hydroxide failed to induce CMI [117]. However, in another study, stimulation of CMI was observed following immunization of ponies with Duvaxyn IE-T Plus, an inactivated whole virus EI vaccine adjuvanted with aluminum hydroxide [119]. The uncertainty surrounding CMI stimulation by inactivated whole virus EI vaccine adjuvanted with aluminum hydroxide has necessitated the evaluation of new adjuvants to augment protective immunity with this model of EI vaccine, and squalene-based adjuvants are now in use.

Inactivated whole virus EI vaccines have been reported to elicit immune responses to both the antigenically variable regions on the viral surface glycoproteins (HA and NA) as well as the more conserved viral proteins such as the NP and M1 matrix protein, believed to induce some level of cross-protection [99,123]. More recently, Pavulraj and coworkers demonstrated that immunization of horses with Calvenza-03 EIV/EHV^®^, an inactivated EI vaccine containing Carbimmune^®^ as an adjuvant, effectively stimulates protective EIV-specific humoral and CMI responses in vaccinated horses [124]. Specifically, the authors showed that a booster immunization of horses with Calvenza-03 EIV/EHV^®^ maintained protective antibodies as measured by single radial hemolysis (SRH) for 6 months in vaccinates that received homologous first vaccine dose and 10 weeks when horses received a heterologous first vaccine dose. They also observed a significant increase in EIV-induced interferon-gamma (IFN-γ)-secreting peripheral blood mononuclear cells (PBMC), a marker of CMI in immunized or infected horses, in Calvenza-03 EIV/EHV^®^-immunized horses, which was attributable to the presence of Carbimmune^®^ adjuvant in the vaccine. Taken together, a potent inactivated whole virus EI vaccine should contain an adjuvant that can prime the induction of not only a robust, long-lived NAb response but also a strong CMI response that is capable of offering cross-protection against antigenically different strains of EIV.

## 5. Subunit EI Vaccines

Unlike inactivated whole virus EI vaccines, subunit EI vaccine technology incorporates only EIV antigenic fragments (HA or NA) to elicit a protective immunity instead of the entire virus. These antigenic fragments of EIV are then encapsulated in colloidal particles as an antigen delivery system to the host immune system. Of the numerous vesicles available for presenting antigenic components to the host immune system [125], the use of the immunostimulating complexes (ISCOMs) and ISCOMATRIX^TM^ are predominant. ISCOMs and ISCOMATRIX^TM^ possess an in-built adjuvant (Quil A saponin) and, consequently, have been found to be more immunogenic than other colloidal systems such as liposomes and protein micelles [126]. ISCOMs were first described by Morein and coworkers in 1984 as a novel structure for antigenic presentation of membrane proteins from enveloped viruses with potent immunogenic activity [127]. ISCOMs are particulate antigen delivery systems composed of antigen, cholesterol, phospholipid, and Quil A saponin [128], while ISCOMATRIX^TM^ has essentially similar composition but with the antigenic component combined at a later step to create the vaccine [129]. The antigen uptake, processing, and presentation by antigen presenting cells (APCs) is believed to be similar after immunization with ISCOMATRIX™ or ISCOM-based vaccines [126,130,131]. Antigens adjuvanted to ISCOMs/ISCOMATRIX^TM^ are processed by both endogenous and exogenous pathways and presented to major histocompatibility complex class I (MHC I) and MHC II molecules, respectively [132,133]. Many studies have demonstrated the safety and ability of ISCOMs/ISCOMATRIX^TM^ to induce strong antigen-specific humoral and CMI to a wide array of antigens in different animal models [134,135,136,137]. The overall adjuvant activity of the Quil A saponin moiety of ISCOMs/ISCOMATRIX^TM^ has been approved for veterinary use [134,138], but their use in human vaccines has been restricted due to undesirable side effects, such as local reactions, hemolytic activity, and systemic toxicity [138].

The immunity induced by conventional inactivated and subunit EI vaccines in the horse is usually short-lived, primarily an antibody-based response. This has necessitated recent vaccine strategies to aim at the possibility of inducing robust EIV-specific antibody- and T-cell-driven long-term immune responses. Several studies have investigated the efficacy and safety profiles, duration of immunity, immunity gap, and the level of protective immunity evoked by ISCOM/ISCOMATRIX^TM^ vaccine in horses following experimental challenge infection with a wild-type (WT) EIV [104,105,108,139,140,141,142,143,144,145,146]. While most of these studies investigated T_H_2 antibody-based response, studies focused on T_H_1-mediated immune protection following vaccination with EIV-ISCOM/ISCOMATRIX^TM^ in horses are limited. In one study, Paillot and coworkers investigated the EIV-specific protective immune responses in ponies vaccinated with an ISCOM-EIV vaccine (EQUIP F) containing both EIV H7N7 (A/eq/Newmarket/77) and H3N8 (A/eq/Borlänge/91 and A/eq/Kentucky/98) strains before an experimental challenge with A/eq/South Africa/4/03 [146]. Findings from their study showed that the subunit vaccine induced a balanced T_H_1/T_H_2 immune response mediated by increased antibody as measured by SRH and EIV-specific IFN-γ synthesis that correlated with reduced clinical severity and virus shedding in vaccinated ponies compared to the controls. Further studies are needed to fully characterize the cell-mediated responses of horses immunized with EIV-ISCOM/ISCOMATRIX^TM^ and their roles in immune protection against EI.

## 6. Recombinant Virus Vector Vaccines

For decades, scientists have been researching approaches for introducing a small immunogenic segment of an infectious virus into a safe, larger-coding virus in an attempt to induce a long-term protective immunity against the foreign gene in the target host upon immunization [147,148,149]. Viral vectors are viruses that can be genetically modified to encode gene segments of another virus that is immunogenic for the purpose of vaccine production. Viral vectors have the potential for applications in both gene therapy and vaccine production. The concept of recombinant virus vector vaccine is different from that of inactivated whole virus and subunit EI vaccines that predominantly stimulate the humoral arm of the host adaptive immune response. Recombinant viral vector vaccines deliver antigens to the intracellular compartment in their targets, stimulating a robust and long-lasting CTL response in the process, leading to the elimination of virus-infected cells [150]. The development of viral vectors is a great biological safety concern, and selected viral vectors for vaccine development have to be proven safe to gain public acceptance. As a result, most viral vectors are replication-deficient and, in most cases, genetically engineered to eliminate pathogenicity, making them unable to cause any harmful effect in immunized individuals. Experimentally, vaccinia virus and adenovirus are two of the most commonly used viral vectors because of their capacities for gene insertion, capability of attenuation in mammalian cells, and their ability to induce robust CMI, specifically, the T_H_1-directed response against the expressed foreign antigen following immunization.

A robust protective immunity, comprising EIV-specific antibody, measured by SRH, and IFN-γ+ cells, was reported in ponies vaccinated twice (at 35-day interval) with a licensed ProteqFlu^®^ canarypox vector vaccine (Boehringer Ingelheim, Ingelheim, Germany)), expressing the HA genes of influenza A/eq/Newmarket/2/93 and A/eq/Kentucky/94 (H3N8), following an experimental challenge with a representative circulating equine A/eq/Newmarket/5/03 strain [151]. In a similar study, ponies that were immunized with a canarypox vector vaccine with the same antigenic composition demonstrated an early onset of induction of anti-EIV antibody response with protection that lasted 6 months following challenge with equine A/eq/Kentucky/91 H3N8 or A/equine/Ohio/03 H3N8 [152]. Although the HA of IAV is not a major CTL target, a strong EIV-specific HA-antibody induced by canarypox vector vaccines offers the first line of an adaptive defense to block viral infection, especially with a homologous virus strain. The ProteqFlu^®^ canarypox-vector vaccine, by virtue of expressing only HA, is also the only currently licensed DIVA-capable vaccine, and for that reason, was chosen for exclusive use in Australia to contain their 2007 EIV epizootic [11]. In situations where the circulating field strain of EIV has evolved and undergone significant mutation in their HA epitopes, this HA-based approach may offer little or no protection. However, the NP of IAV are a major target of immunodominant CD8+ T-cell responses [153], and immunization of ponies with a modified vaccinia Ankara vector vaccine expressing NP of A/equine/Kentucky/1/81 H3N8 (Eq/Ky) induced high levels of EIV-specific serum IgGa and IgGb as well as virus-specific lymphoproliferative responses in all vaccinated ponies [154]. Thus, a recombinant viral vector vaccine expressing any EIV protein that contains immunodominant CTL epitopes, such as the M1 and NP, would offer a more improved alternative to strengthen both the EIV-specific antibody and T-cell responses already provided by HA-expressing viral vector vaccines. Further work is required to investigate immunogenic CTL epitope-rich regions within the most antigenic viral proteins of EIV that can be co-expressed in recombinant viral vector systems for T-cell-mediated multistrain EI vaccine development.

A presumed limitation to the use of viral vector vaccines is the presence of pre-existing immunity to the vector, itself. For instance, the acquisition of immunity against the poxvirus component of vaccinia-vectored vaccine may reduce the vaccine’s effectiveness in later vaccinations [155]. Vaccinia virus presents human safety issues, and efforts are being continuously made to improve vaccinia-based virus vectors. The abortive infection produced by canarypox, an avipoxvirus, reduces the vector-directed immune response as well as enhances the safety of canarypox-vectored vaccines [156]. For the horse, the canarypox virus vector vaccine appears to be well-tolerated in the host for immunization against West Nile virus [157,158,159]. Other potential vectors, e.g., equine adenovirus, present opportunities for development. There are also prospects for indirect application of viral vector technologies, e.g., plant production of artificial virus-like particles (VLPs) containing influenza antigens to be used as immunogens [160].

## 7. DNA-Based EI Vaccines

Recombinant DNA technology has made important contributions to both human and animal health, especially in the field of vaccinology. This vaccine strategy involves the injection of plasmid DNA encoding an antigen into host cells, where it can undergo translation, activating the CD4+ and CD8+ cells for an enhanced antigen-specific antibody and CTL response. Like the recombinant-vector vaccine, this vaccine technology is advantaged compared to the inactivated-virus and subunit systems, because it allows for the de novo synthesis of the immunogen, over a period of time, within the host cell following the injection of the DNA plasmid. This allows for the antigen presentation in its most native conformation [161,162] and the appropriate expression of both MHC class I and MHC class II molecules [163,164] similar to that in natural virus infection but devoid of risk of reversal to virulence, as with a live virus [165]. The DNA technology facilitates rapid updating by site-directed mutagenesis, which is particularly relevant for influenza vaccines. Messenger RNA-based vaccines work on the same principle and, as this technology has advanced for human applications, including their recent use to combat Severe Acute Respiratory Syndrome Coronavirus 2 (SARS-CoV-2), the etiological agent of coronavirus disease 2019 (COVID-19), it may soon come into veterinary applications.

The application of DNA technology to EI vaccines has been ongoing experimentally for some time, and it is gradually gaining acceptance as a possible basis for new generation vaccines in horses. In a study by Ault and colleagues, the ability of a recombinant DNA vaccine expressing HA antigen(s) from EIV H3N8 strains, either in a monovalent or multivalent composition, was assessed for its ability to induce protective immune responses against homologous EIV challenge in vaccinated ponies. Ponies were either immunized with a monovalent DNA expressing the HA gene of A/Equine/Ohio/03 H3N8 or a trivalent DNA mix of HAs of A/Equine/Ohio/03 (H3N8), A/Equine/Bari/05 (H3N8), and A/Equine/Aboyne/05 (H3N8) before challenge with A/Equine/Ohio/03 (H3N8) WT. All vaccinated ponies showed a similarly elevated EIV-specific antibodies, as measured by hemagglutination inhibition (HI) and SRH, which conferred protection against clinical disease and nasal shedding of virus compared to the sham-vaccinated controls [166]. The magnitude of this response was similar to that reported for both modified-live and canarypox-vectored vaccines but was of a slow onset. Pioneering works in the field have previously demonstrated that there is a benefit to extending the duration between the prime and booster doses of DNA vaccine expressing HA of EIV in mice for a stronger protective immune response [167]. Therefore, the success of a DNA-based vaccine for the control of EI would depend on the optimization of the time between the initial and booster DNA vaccinations for a more potent effect in vaccinated horses.

The characteristics of the induced immune response in ponies that received the HA DNA vaccination indicate increases in EIV-specific IgGa and IgGb antibody [168,169] and IFN-γ [169] responses, which are correlates of protection from EI, even in the absence of local mucosal IgA [168]. Although HA-based DNA vaccines were able to offer protection from clinical disease despite the lack of IgA stimulation, induction of mucosal IgA has a desirable effect in the control of influenza virus before it enters systemic circulation [170]. In addition, the induction of virus-neutralizing IgA represents a valuable correlate of cross-protection against antigenically different IAV [171], and during transport across the mucosal epithelium, IgA may also help to neutralize intracellular viruses [172]. Although the immunization of ponies with HA-based DNA vaccines clinically protects from EI due to the enhancement of adaptive immune responses other than IgA, factors that favor the induction of IgA in the respiratory mucosa should be encouraged during the development of DNA vaccines. This will go a long way to prevent nasal shedding of the virus and help to curtail the possibility of EIV transmission to naïve horse populations.

## 8. Modified Live-Attenuated Vaccines (MLV)

This technology involves the manipulation of a live virus under laboratory conditions such that it can no longer cause harmful effects but retains its immunogenicity upon immunization. MLV can induce strong, long-lasting cross-protection by eliciting local mucosal immunity as well as systemic B- and T-cell proliferation. As whole intact viruses, they also present CTL epitopes on NP/M proteins which should facilitate cross-protection against heterologous WT strains. MLV mimics a live IAV in vivo following administration. They are recognized and presented to the APC like a live virus capable of continuous antigenic stimulation of the host’s mucosal and systemic immune systems. This helps to facilitate memory cell production. Unlike the conventional killed virus vaccines, an MLV does not need to be potentiated by adjuvants to induce a robust and prolonged protective immunity in horses. During development of an MLV, the level of attenuation should be fine-tuned such that virus is able to evoke protective immunity in the host while still maintaining safety. As a result, the most important consideration for an MLV is safety, especially in immunocompromised individuals. The attenuation must be safe and not able to revert to virulence for public acceptance. To further minimize the possibility of reversion, the live vaccine organism should be self-limiting and incapable of spreading spontaneously from vaccinated to nonvaccinated animals [173]. Due to these safety concerns, and with the example of human polio in mind, some countries forbid MLV veterinary vaccines, all together. The two commonly used approach for the generation of an EI MLV are the cold-adaption (ca), rendering the virus incapable of replication at normal host body temperature, and the reverse genetic system.

Live-attenuated influenza vaccine (LAIV) based on the generation of temperature sensitive (ts), ca, strains of EIV were first tested by Holmes et al. [174,175]. Flu Avert^®^ I.N. Vaccine (Merck Animal Health Inc., New Jersey, USA), the first modified-live virus, intranasal EI LAIV, was licensed in 2000 and remains on the market in the US. This vaccine was shown to be safe, phenotypically stable, and effective in protecting vaccinated ponies against either homologous or heterologous EI challenge [173,176,177]. It was also safe and effective in ponies with exercise-induced immunosuppression [178]. This clinical protection was achieved despite induction of only low or even undetectable serum antibody titers, suggesting that the ts/ca virus induced mucosal immunity directly in the respiratory tract. This vaccine was ca by serial passage of the WT influenza A/equine//Kentucky/91 virus in embryonated eggs with step-wise reduction of the incubation temperature to 26 °C [179]. Consequently, the attenuated virus can replicate in the upper respiratory tract (URT) of horses where immune responses are induced but not in the warmer environment of the lower respiratory tract (LRT) of the host [180]. This MLV vaccine had the limitation that a single dose reduced but did not eliminate WT virus shedding following challenge: i.e., it did not provide sterile immunity. In a more recent study, Tabynov and coworkers investigated the duration and protective immunity of a ca, modified-live attenuated EI vaccine in a group of yearlings. Two groups of horses, comprising of a prime group and a booster group, were vaccinated with a ca equine A/HK/Otar/6:2/2010 H3N8, and protection from clinical disease was evaluated, relative to the control group, upon challenge with either homologous or heterologous EIV. The authors demonstrated that the vaccine is safe and offers a protective immunity either in the prime or the booster group, with protective immunity lasting for 12 months, even against heterologous EIV challenge [181]. This vaccine induced EIV-specific HI titers that correlated with reduced clinical and virologic burdens of EI.

## 9. Reverse Genetics to Generate EIV Live-Attenuated Influenza Vaccines (LAIV)

Reverse genetics approaches to generate recombinant IAV, including EIV [182,183], have been well-described in the literature, including those for the development of LAIV [184,185,186,187,188,189] (Figure 6). The most commonly used approach to generate recombinant IAV is based on the use of ambisense plasmids that allow the expression from the same plasmid of the negative-stranded vRNA and the positive-stranded mRNA from transfected cells (Figure 6A). In this rescue system, a cDNA copy of each of the vRNA segments are cloned in an ambisense plasmid containing, in opposite orientation, cellular RNA polymerase I (hPol-I) and RNA polymerase II (Pol II) cassettes (Figure 6A). The RNA Pol I cassette directs, under a hPol-I promoter and a murine terminator sequences, the synthesis of vRNAs. In opposite orientation, the RNA pol II cassette directs, under a Pol-II promoter and the bovine growth hormone (BGH) polyadenylation signal, the synthesis of viral proteins required to initiate viral genome replication and gene transcription (Figure 6A). Once the eight viral segments are cloned in the ambisense plasmid, the eight plasmids are cotransfected into cocultures of human 293T and canine Madin–Darby Canine Kidney (MDCK) cells and rescued recombinant IAV are amplified in either MDCK cells or chicken embryonated eggs from the tissue culture supernatant from transfected cells (Figure 6B).

## 10. EIV LAIVs Based on Truncations of the Viral NS1 Protein

IAV NS1 is a multifunctional protein with several roles during viral replication [70,190,191,192,193,194,195]. One of the most important and characterized functions of IAV NS1 is to counteract the antiviral innate immune response induced during viral infection [70,194,195]. EIV NS1, similar to other IAV, is able to efficiently counteract the interferon (IFN) response [196]. Recombinant IAVs with deletions, truncations, or modified NS1 proteins have been shown to represent an excellent option for their implementation as LAIV since they are not able to replicate in IFN-competent hosts due to the lack or altered anti-IFN function of NS1. The use of IAV with modified NS1 proteins as LAIV has been proposed for swine [197,198,199,200], avian [201,202,203,204,205], canine [206], human [207,208,209], and equine [182] IAV based on their safety, immunogenicity, and protective/effective profile. Importantly, a correlation between the length of IAV NS1 protein and attenuation, immunogenicity, and protection efficacy in vivo has been reported. Likewise, recombinant influenza B virus (IBV) containing a deletion and/or truncations of the viral NS1 protein has also been proposed as LAIV for the treatment of IBV infections [210].

Using reverse genetics for influenza A/eq/KY/5/02 H3N8, Quinlivan et al. generated recombinant EIV containing 1–73, 1–99, and 1–126 amino-acid-long NS1 proteins (Figure 6C) that were tested for safety in mice [182]. Contrary to previous studies with other IAV, Quinlivan et al. found that, in the case of EIV, the length of the NS1 protein did not correlate with the level of attenuation [182]. In the EIV strain used in this study, the virus with the shortest NS1 protein [1,2,3,4,5,6,7,8,9,10,11,12,13,14,15,16,17,18,19,20,21,22,23,24,25,26,27,28,29,30,36,37,38,39,40,41,42,43,44,45,46,47,48,49,50,73,74,75,76,77,78,79,80,81,82,83,84,85,86,87,88,89,90,91,92,93,94,95,96,97,98,99,100] was less attenuated than those encoding larger NS1 proteins (e.g., 1–99 and 1–126) [182]. Importantly, vaccination in horses demonstrated that aerosol or intranasal vaccination with the NS1 truncated EIV A/eq/KY/5/02 H3N8 were safe, and horses vaccinated with the EIV NS1–126 A/eq/KY/5/02 H3N8 were protected against developing fever and clinical signs of infection and experienced reduced quantities of challenge A/eq/KY/5/02 H3N8 EIV WT compared to mock-vaccinated controls, suggesting the feasibility of implementing these NS1 truncated EIV as safe, immunogenic, and protective LAIV against EI in its natural host [211].

## 11. LAIVs Based on ts, ca, and att EIV

In 2018, Rodriguez et al. described the generation of an EIV LAIV for the FC-1 Ohio/03 H3N8 [183]. This clade 1 EIV LAIV was generated by introducing the mutations responsible for the ts, ca, and attenuated (att) phenotype of the human master donor virus (MDV) A/Ann Arbor/6/60 H2N2 [212,213,214,215] into the polymerase PB2 (N265S) and PB1 (K391E, E581G, and A661T) segments of EIV Ohio/03 (Figure 6D). The mutations of the human MDV A/Ann Arbor/6/60 H2N2 were able to transfer the same ts, ca, and att phenotype to the EIV Ohio/03 [183]. When tested in mice, the Ohio/03 LAIV was safe, immunogenic, and able to protect, upon a single intranasal dose, against Ohio/03 WT infection [183]. Importantly, its safety, immunogenicity, and protective efficacy profile was also observed in horses, suggesting the feasibility of implementing this LAIV for the control of FC-1 EIV in horses [183]. Since the OIE currently recommends that both FC-1 and FC-2 EIV should be incorporated in any EIV vaccine, a year later, Blanco-Lobo et al. developed a FC-2 monovalent EIV LAIV and, also, the first bivalent EIV LAIV for the protection against FC-1 and FC-2 EIV based on the use of reverse genetics approaches [216]. To that end, Blanco-Lobo et al. used the internal genes of the Ohio/03 LAIV as a MDV [183] and the HA and NA of A/equine/Richmond/1/2007 H3N8 (Rich/07), a representative FC-2 EIV, to generate the FC-2 EIV LAIV [216]. To generate the bivalent EIV LAIV, Blanco-Lobo et al. used a previously described approach to generate a bivalent canine influenza virus (CIV) LAIV [217] that consisted of mixing the two FC-1 and FC-2 monovalent EIV LAIV and demonstrate how vaccination of horses with the bivalent EIV LAIV was safe and able to protect, upon a single intranasal dose administration, against challenge with both FC-1 A/equine/Kentucky/1/1991 H3N8 and FC-2 Rich/07 H3N8 EIV in horses [216].

## 12. Vaccine Efficacy Testing

The first generations of EI vaccines were killed virus preparations that included both the equine/H7N7 and equine/H3N8 subtypes. Efficacy was assessed by seroconversion of horses following vaccination [218]. Typically, such testing is done by inoculation of cohorts of horses (suitably blocked for factors including breed, age, sex, prevaccination titers) with one dose, or two doses usually 1 month apart, and blood collection at 1- or 2-week intervals. Peak response is typically 2–3 weeks after each dose. Duration-of-immunity studies feature successive blood collections for 6–12 months. A low-titer sentinel cohort should be maintained with each vaccination cohort to alert if there was incursion of WT EIV in the study population. With conventional killed-virus vaccines, inoculation is by intramuscular injection into the horse’s neck muscles. 

Seroconversion is widely assessed by using the HI test [219]. HI testing is known for a deficiency of reproducibility, particularly interlaboratory reproducibility [220]. Alongside mechanical error, two other factors are recognized: First, sera contain nonspecific inhibitors of hemagglutination. These are of three classes, α, β, and γ, varying in heat-lability and virus neutralization activity. Equine serum, in particular, contains high levels of a heat-stable, virus-neutralizing γ-inhibitor called α-2 macroglobulin [221,222], producing false-positive results. This is removed from sera before HI testing, with different levels of effectiveness, by various treatments including heat (56 °C), receptor-destroying-enzyme (RDE), kaolin adsorption, or trypsin–periodate treatment [223]. Curiously, the nonspecific inhibitor of EIV H7 HA is harder to eliminate than that of EIV H3 HA and requires trypsin–periodate, the most vigorous treatment, to assure an absence of false-positive results when testing for EIV H7 antibodies [224]. For this reason, periodate or RDE are the serum treatments recommended by the OIE for HI testing of equine sera. Second, it was found that the equine/H3 component of vaccines was much less immunogenic than the equine/H7 component. This is believed to be due to more extensive addition of carbohydrate sidechains masking antigenic determinants. Adjuvants such as alum or squalene help overcome this problem, but in general, postvaccination serum antibody titers to the equine/H3 component are unimpressive. For HI testing, a virus antigen pretreatment, Tween80/ether [225], is sometimes adopted [218] that amplifies the apparent antibody titer by disrupting whole virus into lipid micelles. Tween80/ether treatment introduces problems of batch-to-batch variation and lack of homogeneous stability that contribute to the reproducibility problems of HI testing, and there is no reliable conversion factor between titers obtained using ether-treated antigen and those obtained without. The best procedure is to test all samples side-by-side. Furthermore, the increased sensitivity tends to obscure differences in antibody reactivity between virus strains [226]. By HI, seroconversion is conventionally defined as four-fold increase in postvaccination titer over the prevaccination titer. A titer of 1:64 is conventionally viewed as clinically protective, but in horses, this has not been rigorously established, especially against challenge with heterologous strains. It is sometimes erroneously assumed that an HI titer is equivalent to a NAb titer. There is a good correlation [227]; still, the latter should be determined by a neutralization assay, whereas HI is a binding assay [228].

As an alternative to HI testing, the SRH test is often used [229]. This is more reproducible than the HI test [230] and has the further advantage that titers vary linearly by measure of the area of the zone of hemolysis instead of by HI’s successive two-fold increases that require log-transformation for statistical analysis. Using SRH, titer thresholds are more precisely correlated with clinical protection (prevention of disease, 85 mm^2^) or virological protection (prevention of virus shedding, also called sterilizing immunity) against homologous (150 mm^2^) or heterologous (up to 205 mm^2^) virus strains [88,107,231]. The SRH test has its own technical problems (e.g., different viruses may require reoptimization of the relative quantity of erythrocytes), but these optimization issues seem to have little effect on the titers finally obtained. Seroconversion is conventionally defined as an increase of at least 25 mm^2^ or 50%, whichever is smaller, in area of the zone of hemolysis. Other serological assays (reviewed in [232]) are sometimes used, although generally not for efficacy testing. These include virus neutralization [97,227] and enzyme-linked immunosorbent (ELISA) [233] assays.

Once equine antibody isotypes were characterized, it was discovered that conventional killed-virus vaccines adjuvanted with alum mainly induced serum antibodies of subclass Ig(T) [234], i.e., IgG3/5 in the modern nomenclature [235]. This differed from natural EIV infection of horses, which induced mainly antibodies of the IgGa/b (IgG1/4/7) subclasses; furthermore, Ig(T) antibodies were shown to be less protective than IgGa/b antibodies. Additionally, the alum-adjuvanted killed virus vaccine was incapable of generating a mucosal IgA response characteristic of natural infection and was thought to be poor at stimulating cell-mediated immune responses: more recent work suggests that alum adjuvants stimulate T_H_2 responses but inhibit T_H_1 responses (reviewed in [116]). These factors offered partial explanations for the anecdotal experience of veterinarians in the early 1990s, that the available EI vaccines were of limited effectiveness, and contributed to the push for the development of vaccine technologies that would provide improved protection: ISCOM, MLV, and recombinant-canarypox vaccines that came to market between 1995 and 2005.

The development of an MLV for EI triggered a re-evaluation of standards for vaccine efficacy testing, because this MLV was administered by the intranasal route and was notably poor at induction of serum HI antibodies. It was predicted that this vaccine would be more effective at induction of mucosal antibodies and cytotoxic T-lymphocytes in the respiratory tract, although this has actually not been conclusively demonstrated. Instead, efficacy was demonstrated by experimental challenge of vaccinated horses with WT EIV and observation of clinical and virological parameters. That is, protective efficacy was demonstrated by actual protection instead of by seroconversion, which is a correlate of protection. In the US, this has become accepted as satisfactory and, indeed, desirable evidence for vaccine license approval when conducted according to US Department of Agriculture (USDA) requirements.

Experimental challenges of horses have their own problems: (1) only well-isolated or purpose-bred herds are likely to have no exposure to EI, and when challenged, even weakly seropositive horses may show evidence of protection without vaccination; (2) horses are not genetically homogeneous. While all equids appear to be susceptible to EI, it is possible that some breeds (donkeys) are more severely clinically affected than others; (3) some clinical signs can be measured with reasonable accuracy, e.g., body temperature and heart rate. Others are more subjective, e.g., lung auscultation sounds, demeanor, severity of nasal discharge. A large-animal veterinarian, preferably blinded to vaccination status, should be part of the research team, to make these judgements and provide therapies, if needed; (4) virological assessment is based on virus content detected in nasal or (ideally) nasopharyngeal swabs. Horses frequently object to the swabbing procedure with the result that the quality of the swab sample may vary from horse to horse and day to day, affecting the accuracy of virus quantitation; and (5) the ideal solution to these problems is to have large numbers of horses in vaccine and control groups. That is expensive and likely to be limited by the stall capacity of the animal facility. The outcome is that the statistical power of such experiments may be less than desired. A successful challenge is one where the clinical and virological differences between vaccinates and controls is unequivocal. To overcome these problems, small animal models for testing EI vaccine efficacy have been developed (e.g., [236,237]), but testing in the actual target species of interest remains desirable.

The procedure for experimental challenge of horses is that described by Mumford et al. [238]. Those authors showed that aerosol inhalation was more effective than intranasal intubation as the route of virus administration and that a dose of 10^6^ 50% egg infective dose (EID_50_) units was needed for full expression of typical clinical signs. This is presumably far greater than the natural infection dose, arguing that some virulence for horses may be lost by egg adaptation. Using this or similar protocols [239], influenza-naïve horses will typically exhibit pyrexia, coughing, and serous nasal discharge by the second day following infection. Virus shedding is sometimes detectable on the first day following infection and typically reaches its peak on the second or third days. Shedding then declines, and live virus may become undetectable by days 6 to 8 following infection, while vRNA detectable by qRT-PCR may persist for 10 days or even longer. There have been reports of RT-PCR-positive swabs beyond day 15 [240], although these horses are presumably no longer contagious. Severity of clinical signs—pyrexia, tachypnea, tachycardia, cough, mucopurulent nasal discharge, anorexia—may peak around days 2–3, or there may be a second peak around days 4–5 triggered by secondary bacterial infections. Swelling of submandibular or retropharyngeal lymph nodes may be detected, but in our opinion, this is an outcome of the immune response and may be apparent in otherwise clinically protected horses. Generally, horses will have completely recovered from disease signs and be clinically normal by day 10. There may be considerable horse-to-horse variation. In particularly severe cases, there may be bronchopneumonia with coarse crackles detected by lung auscultation, and it is possible for bronchopneumonia to become life-threatening without antibiotic treatment. In experimental settings, animal welfare considerations may demand antibiotic therapy before clinical signs become so severe. Some laboratories instigate prophylactic antibiotic treatment prior to challenge to avoid secondary infections, all together. Along with virus shedding quantified by qRT-PCR [241] or by EID_50_ assay, measurable clinical signs such as pyrexia and tachycardia can be statistically analyzed. To quantify the more subjective clinical signs for statistical analysis, our laboratories use an established but essentially arbitrary scoring system shown in Table 5 [173]. Alternative scoring systems can be envisioned. Baseline data for each horse should be collected in the days immediately preceding challenge, after horses have become settled inside the facility. Once horses have had clinical interventions postchallenge, e.g., antibiotic, anti-inflammatory, or other therapies, their subsequent clinical scores are dropped from the analysis. Typically, serum antibody titers of both vaccinated and control horses increase after challenge. EIV vaccines must also be tested for sterility, purity, freedom from extraneous agents, batch potency, and safety, i.e., absence of abnormal local or systemic reactions in the recipient horse following vaccination. For inactivated-virus vaccines, potency standardization is typically done by single radial diffusion assay [242].

## 13. Vaccine Updating

As described elsewhere, EIV strains in circulation undergo antigenic drift, albeit at a much slower rate than human, avian, or swine IAV. Therefore, vaccine manufacturers, to keep up with antigenic drift, should periodically update the virus strains in their vaccines. However, vaccine manufacturers are not required to do so, and it is possible for a product to remain on the market long after it has become antigenically obsolete. Leading manufacturers understand the scientific need for updating, but prior to the 1990s, there was no guidance, as the evolution of EIV H3N8 HA at the molecular level was just beginning to be understood. In the wake of large outbreaks of EIV in Europe in 1989 and in Hong Kong in 1992, Dr. Jennifer A. Mumford (Animal Health Trust, Newmarket, UK) instigated a series of meetings that for the first time brought together virologists from around the world to address the problem [243]. Those meetings led to the realizations that there were two cocirculating lineages of EIV H3N8 HA that had significantly diverged from the vaccine virus strains in use (mainly Fontainebleau/79, Kentucky/81) [85], and there had been no confirmed isolations of equine H7N7 virus since 1979 [81]. The meetings further led in 1995 to the founding of the international OIE/World Health Organization (WHO) Equine Influenza Expert Surveillance Panel (hereafter ESP). This consists of the heads of the OIE’s international reference laboratories for EI, one or more representatives of WHO Collaborating Centers for influenza, heads of national or regional EIV surveillance operations (e.g., [94,244]), and veterinary virologists who actively study EI in regions including South America and east Asia. The role of the ESP is to serve as an advisory body to the equine industry regarding the updating of vaccine virus strains. Its members collate information on regional EIV outbreaks, the virus strains responsible for them, and vaccination status of affected horses. Member laboratories perform virus characterizations including nucleotide sequencing and antigenic comparison using reference antisera, which information feed into antigenic cartography analysis. On these bases, the panel, meeting annually, debates whether the vaccine strains are still a good antigenic match for the virus strains in current circulation and, if not, then what the updated vaccine strains should be. The OIE publishes the ESP’s annual conclusions and recommendations in its Bulletin (online at oiebulletin.com). Both the European Medicines Agency [245] and the USDA [246] have EI vaccine updating guidelines that recognize the guidance of the OIE–ESP recommendations.

The technique of antigenic cartography [247] has become a critical component of the evaluation of antigenic drift in EIV isolates. This technique, involving multidimensional correlation of masses of HI data comparing new vs. old virus strains against a panel of reference ferret antisera, demonstrated the antigenic divergence of the H3N8 EIV into the “Eurasian” lineage and the “American” FC-1 and FC-2 sublineages [248]. These each form distinguishably separate antigenic clusters. The ESP has followed a policy that, so long as new isolates continue to map to the existing FC-1 and FC-2 clusters, i.e., are antigenically indistinguishable from the clade prototype strains, the standing vaccine virus strain recommendations for those clades remain prima facie valid. This has produced the surprising outcome that, as late as 2021, the recommended vaccine virus strains remain influenza A/equine/South Africa/2003-like virus (e.g., Ohio/03) for FC-1 and influenza A/equine/Richmond/2007-like virus for FC-2. That is to say, 18 years of antigenic drift has not yet caused the replacement of the FC-1 cluster in the actively circulating EIV, which these authors would not have predicted. Recent large outbreaks in Europe and the UK in 2018–2019, which initially raised concerns about possible emergence of an antigenically new variant, were instead the outcome of the reintroduction of FC-1 EIV into a region where it had been long absent [14]. This was supported by comparative serum neutralization test as well as by HI results [249]. It must be noted that current EIV antigenic cartography is focused only upon HA, because HI data is its resource, and while in principle it could be applied to NA, as well, there is no data for it to analyze. Sequence information shows that EIV NA is also subject to antigenic drift and has established clades that mirror the FC-1 and FC-2 lineages of HA [80], but the significance for EI vaccines of antigenic drift in NA is poorly understood. Serum neutralization testing using pseudotyped-EIV is one approach to overcome this limitation [232,250]. Comparative serum binding to a purified EIV NA protein antigen by enzyme-linked lectin assay (ELLA) [251]) is being investigated.

With regard to the equine H7N7 subtype, the ESP concluded in 1999 that it was no longer relevant and should be dropped from EI vaccines. This has been done in most current vaccines. Occasionally, there have arisen reports suggesting that this subtype may still survive in equids (e.g., [252,253]), but the ESP has not obtained any virus isolate in support of these claims. Part of the problem may be that, for EIV H7N7 serological testing, nonspecific inhibitors of hemagglutination require more stringent removal (i.e., trypsin–periodate) to eliminate false-positive results, as discussed above [224]. Diagnostic testing must still include testing for the EIV H7N7 on the chance, however unlikely, that it may still survive in some hidden natural reservoir. Various strains of EIV H7N7 do survive in storage in multiple laboratories. That leads to a final confounding factor, the reappearance in the field of anachronistic H3N8 EIV strains, termed “frozen evolution” (e.g., [86,254]). These strains, which usually resemble old vaccine strains, bear HA, whose nucleotide sequence would phylogenetically date them as 5–20 years older than their date of isolation. Thus, these are analogous to the reemergence of human H1N1 IAV in 1977. Fortunately, the anachronistic EIV strains have never yet persisted. The presumption is that these strains escaped from frozen storage.

This ESP system is fundamentally similar to the WHO system for updating of human influenza vaccines. The important difference is scale: the WHO human influenza system has probably 100-fold greater input of virus characterization data and at least 10-fold greater geographical comprehensiveness. The WHO system operates on a tight schedule, which it can do by taking advantage of the seasonality of “normal” human influenza, and which it must do because an update of at least one vaccine strain is needed almost annually. EI vaccine strain updates have been much less frequent, and so far, the existing ESP system has been adequate in scale. Geographical coverage has improved since 1995 through its association with OIE.

Swine and canine IAV are not analogous. In the US, swine do travel from one exhibition to another and carry swine influenza virus (SIV) with them (e.g., [255,256]); but swine do not travel internationally, and therefore, swine influenza is not a listed disease in the OIE Terrestrial Animal Health Code. There is a swine influenza branch of the OIE-FAO animal influenza network (OFFLU), but neither it nor any other international body is tasked with making vaccine strain recommendations; that is left to national-level organizations, and there is a need for heightened SIV surveillance in much of the world [257]. The European Union in 2001 instigated a series of SIV surveillance network programs (ESNIP, [258]), that is largely industry-funded. Similarly to EI, swine influenza is not a notifiable disease in the US, but the USDA established a SIV surveillance program through the National Animal Health Laboratory Network in 2009 [257,259]. Swine population structures and management are much different than in equines, with many animals in close contact, often continuously indoors, and a high rate of turnover ensuring a constant supply of juvenile pigs as breeding grounds for porcine respiratory disease complex. Outbreaks are more readily detected, and virological sampling is facilitated by the rope-chewing and snout-wipe methods. Thus, SIV surveillance efforts can generate larger data inputs than the ESP can obtain. SIV also presents a more complex problem than EIV: there are more relevant IAV subtypes, gene constellations, and lineages in cocirculation; there are also high probabilities of reassortment of SIV with human or avian IAV strains; and SIV has zoonotic potential as a human pathogen.

Canines also seldom travel internationally, and CIV is also not a notifiable or transboundary disease. As with SIV, there is no international panel that recommends vaccine CIV strains. A surveillance network exists for CIV in the US, headed by the Animal Health Diagnostic Center at Cornell University, New York. This network currently has 12 partners among the veterinary diagnostic laboratories of different states that have collected >40,000 samples since 2015 with some-3000 positive test results [260].

## 14. Conclusions

EI is one of the most important respiratory diseases of horses. The economic burden of this disease imposed on equine industry and equestrian events runs into billions of dollars [261], and EI also exacts a considerable cost in developing nations where equids are still widely used as working animals. As a result, reinforced efforts aimed at preventing EI outbreaks in horse populations are valuable contributions to the continued survival of large and small equine operations. Vaccination remains the most effective approach for controlling harmful pathogenic viral infections, including EI. In this review, we discussed how the unique biology of EIV facilitates the design of different kinds of vaccines and reviewed data supporting their safety and efficacy in equine subjects, including the pros and cons regarding each vaccine approach. The constant evolution of EIV, transboundary nature of the disease, and evidence of poor vaccine performance have necessitated an ongoing international surveillance effort, unique for animal influenza viruses, advising on the updating of EIV strains incorporated in commercially marketed EI vaccines.

**Challenges:** Despite the availability of different types of vaccines against EI, there are certain challenges to effective immunization in horses against EI identified in this review. These include the constant evolution of the virus, vaccine breakdown, vaccination-induced short-lived immunity, immunity gap, inability of vaccines to induce sterilizing immunity, and, in young horses, interference with maternally derived immunity [110,114,115].

**Future Directions:** Many EI vaccine technologies are used or being investigated for their safety and ability to induce long-lasting protective immunity. So far, it appears that only MLV can induce a long-lasting protective immunity by triggering multiple arms of the immune system. To mimic this, the use of viral proteins with immunodominant epitopes for CD4+ and CD8+ T cell responses as immunogens in other vaccine systems, especially recombinant DNA-based systems, may be beneficial. In addition, the advent of EI new vaccine technologies has facilitated the development of DIVA-capable vaccines. This enables mass vaccination programs against EI without compromising the serological identification of convalescent horses. Considerable practical experience with mRNA vaccines has been gained from their use against SARS-CoV-2, and this technology may soon find veterinary applications against pathogens such as EIV. Many groups are evaluating EI vaccines to understand how to induce long-lasting sterilizing immunity, reduce the effect of strain differences, and close the immunity gap between vaccine doses in young horses. As with human influenza, the prospect of a universal EIV vaccine targeting conserved domains is tantalizingly close.

Since neither EI vaccine coverage nor surveillance are universal, influenza in horses is not going away in the foreseeable future. However, the results from these studies should lead to more improved EI vaccines that will reduce the severity and spread of EI disease outbreaks into minor episodes and also help to combat newly emerging EIV strains in the future.

## Figures and Tables

**Figure 1 viruses-13-01657-f001:**
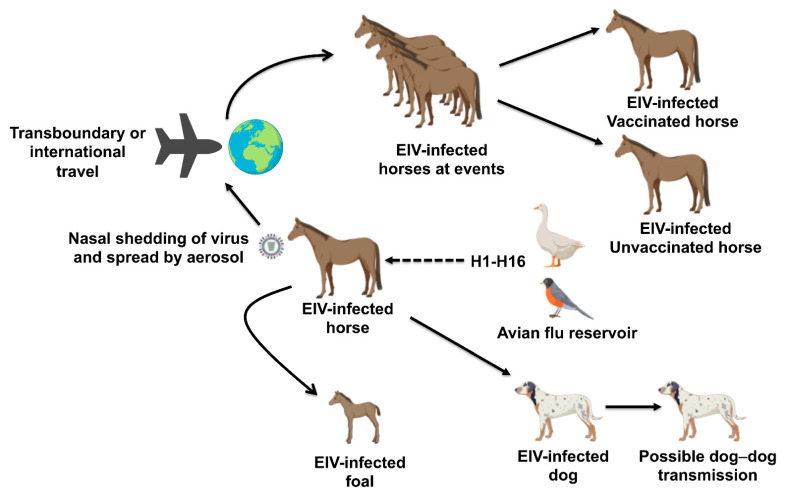
Origin and mode of transmission of EIV: Aquatic birds serve as the natural reservoir of avian influenza, a progenitor of EIV. Following infection of a horse, the virus can be spread to other animals in aerosolized droplets by coughing or through fomite transmission. International transport of horses facilitates the spread of the virus to new geographical locations, especially during horse events such as racing or shows. Dogs can also become infected and transmit the virus to other dogs.

**Figure 2 viruses-13-01657-f002:**
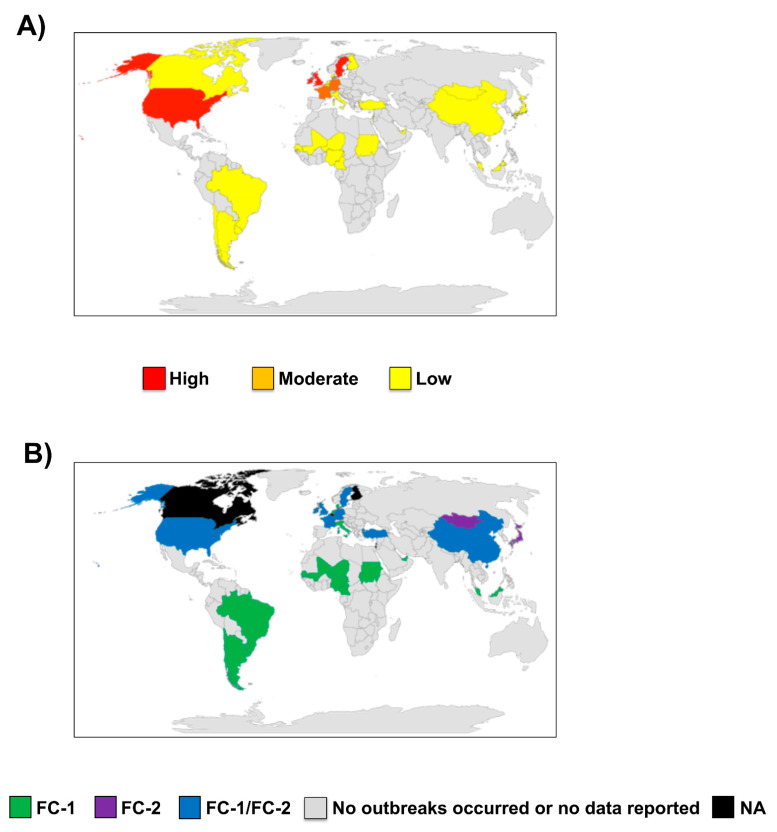
Incidence of EIV in the equine population over the last decade, compiled from OIE ESP on Equine Influenza Vaccine Composition reports, 2010 to 2019. (**A**) Countries are ranked based on the average of incidence rate, i.e., number of years reported to OIE ESP over a period of 10 years. (**B**) Clades responsible for these outbreaks indicated: An incidence rate >50% is classified as high, 50% is moderate, and <50% is low. NA, not available.

**Figure 3 viruses-13-01657-f003:**
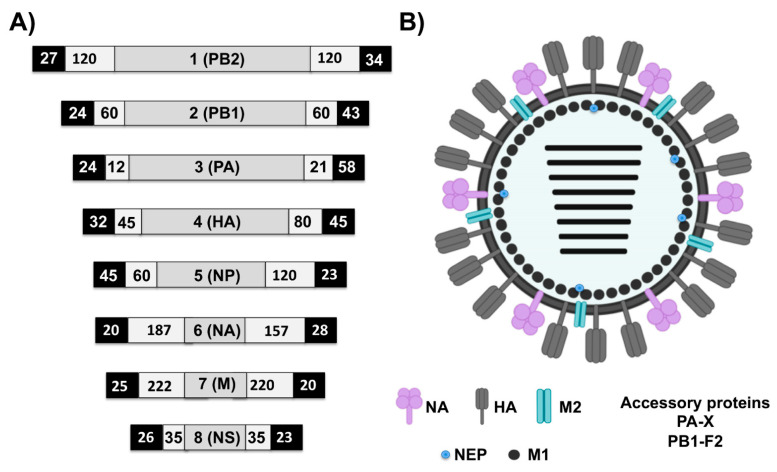
Schematic representation of an influenza A virus (IAV) genome organization and virion structure. (**A**) The genome organization of the eight negative-sense RNA gene segments (PB2, PB1, PA, HA, NP, NA, M, and NS) of IAV. All segments 1—8 (as in Table 2) are numbered according to the conventional representation from 3′ to 5′. The black boxes at the end of each viral segment indicate the noncoding regions (NCR). White boxes show the packaging signals present at the 3’ and 5′ ends of each gene segment. (**B**) The basic architecture of IAV contains envelope surface glycoproteins HA, NA, and M2 ion channel protein, all embedded within the host cell-derived lipid bilayer envelope. Immediately underneath the envelope is a well-organized layer of M1 and NEP proteins surrounding the eight negative-sense viral RNA segments.

**Figure 4 viruses-13-01657-f004:**
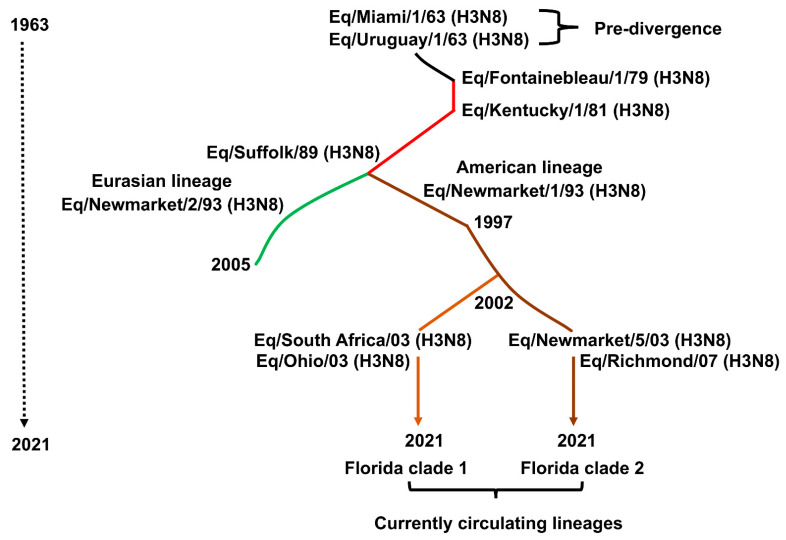
Schematic of evolution of EIV HA, 1963 to 2021. The major lineages as discussed in the text are indicated, as are representative virus strains. Not to scale.

**Figure 5 viruses-13-01657-f005:**
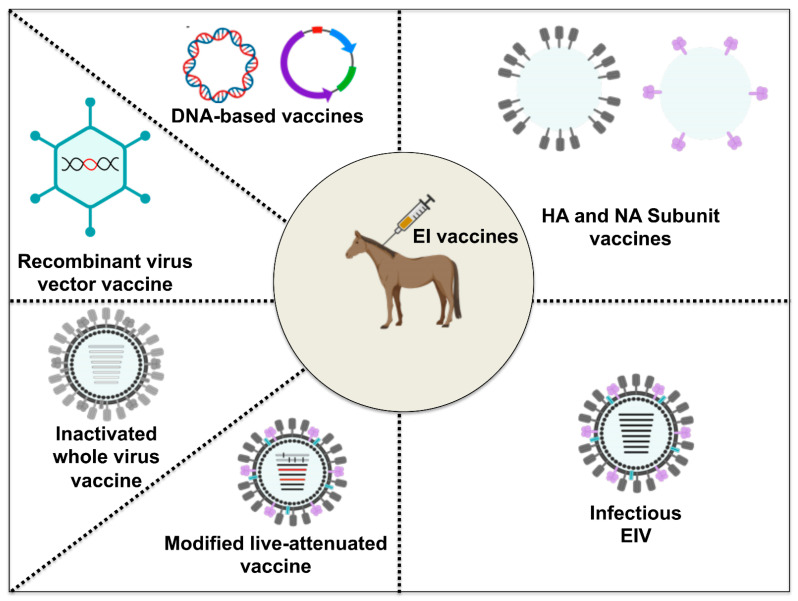
EI vaccine technologies: Each EI vaccine system differs in its antigenic composition and structural organization. These differences impact on the immunogenicity of the vaccine type.

**Figure 6 viruses-13-01657-f006:**
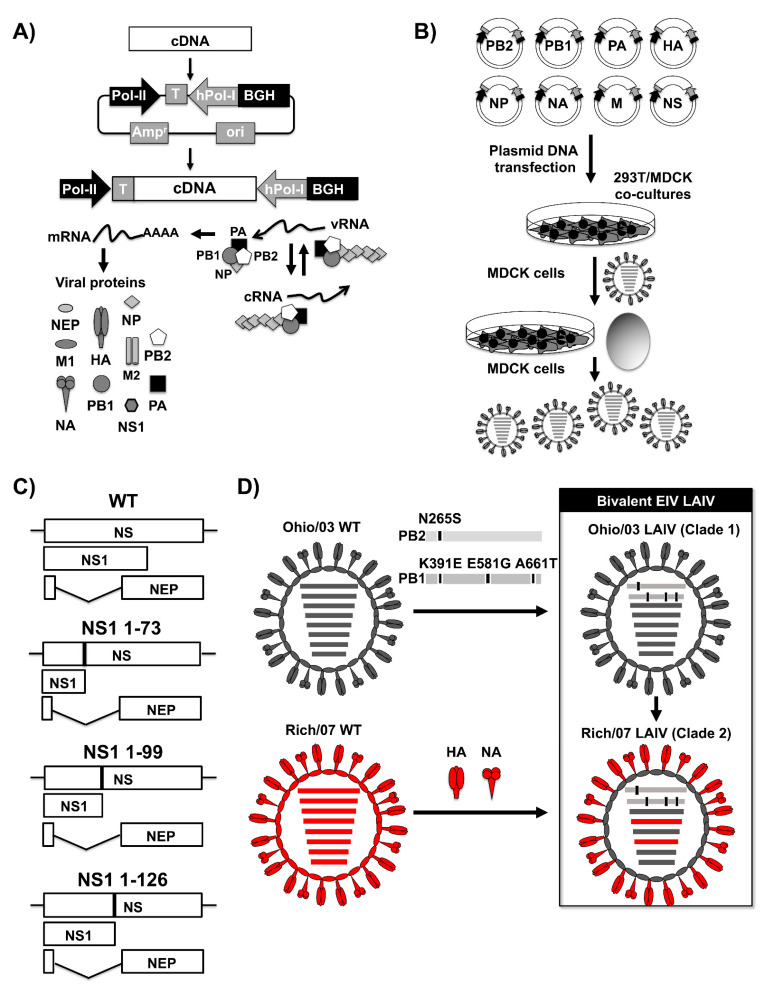
Generation of recombinant EIV LAIV using plasmid-based reverse genetic approaches: (**A**) Schematic illustration of the ambisense plasmids to generate recombinant EIV: Ambisense plasmids for the rescue of recombinant EIV contain the human polymerase I promoter (hPol-I, gray arrow) and the mouse polymerase I terminator (T, gray box) to regulate the expression of the negative-sense vRNAs. In the opposite direction to the Pol-I cassette, the polymerase II-dependent promoter (Pol-II, black arrow) and the bovine growth hormone polyadenylation sequence (BGH, black box) produce viral proteins. The vRNAs generated from the Pol-I cassette are recognized by the EIV polymerase subunits PB2, PB1, and PA that, together with the viral NP, form the viral ribonucleoprotein (vRNP) complexes involved in viral genome replication and gene transcription. Transcription of the vRNA results in viral mRNA and protein production. Replication of vRNAs results in complementary (c)RNAs for the amplification and synthesis of new vRNAs that are incorporated into nascent virions. (**B**) Generation of recombinant EIV using plasmid-based reverse genetics: Cocultures of human 293T and MDCK cells are cotransfected with the eight ambisense plasmids containing the EIV PB2, PB1, PA, HA, NP, NA, M, and NS segments. EIV generated from transfected cells is amplified in fresh monolayers of MDCK cells or in chicken embryonated eggs. (**C**) LAIV based on truncations of EIV NS1 protein: Schematic representation of WT and 1–73, 1–99, and 1–126 NS1 truncated NS viral segments. Black lines represent stop codons introduced to truncate NS1. Lines at the end of the NS segment indicate the 3’and 5’ NCR. (**D**) Schematic representation for the development of an EIV bivalent LAIV: The mutations responsible of the ts, ca, and att phenotype of the human A/Ann Arbor/6/60 H2N2 MDV LAIV were introduced into the PB2 (N265S) and PB1 (K391E, E581G and A661T) viral segments of A/equine/Ohio/1/2003 H3N8 WT (Ohio/03 WT) to generate the FC-1 EIV monovalent LAIV (Ohio/03 LAIV). The backbone of Ohio/03 LAIV was used as an MDV to generate a recombinant virus containing the HA and NA glycoproteins of the FC-2 A/Richmond/1/2007 H3N8 WT (Rich/07 WT) to develop the FC-2 EIV monovalent LAIV (Rich/07 LAIV). The bivalent EIV LAIV was developed by combining the monovalent Ohio/03 and Rich/07 LAIVs.

**Table 1 viruses-13-01657-t001:** Selected EI outbreaks in the past decade.

Country	Year	Number of Animals	Infected Animal	Source
**China**	2010/11	NA	Horse	[31]
**Mongolia**	2010/11	NA	Horse	[31]
**Algeria**	2011	900	Horse	[18]
**Argentina**	2012	NA	Horse	[22]
**Brazil**	2012	NA	Horse	[22]
**Chile**	2012	NA	Horse	[22]
**China**	2012	NA	Horse	[12]
**Kazakhstan**	2012	NA	Horse	[31]
**Uruguay**	2012	NA	Horse	[22]
**Ireland**	2014	118	Horse	[32]
**Malaysia**	2015	25	Horse	[33]
**Brazil**	2015	23	Horse	[21]
**China**	2017	NA	Donkey	[34]
**Uruguay**	2018	NA	Horse	[24]
**Chile**	2018	NA	Horse	[24]
**Argentina**	2018	NA	Horse	[26]
**Senegal**	2019	NA	Horse and donkey	[17]
**United Kingdom**	2019	NA	Horse	[35]
**China**	2020	NA	Donkey	[29]

NA, not applicable.

**Table 2 viruses-13-01657-t002:** EIV segments and their functions.

EIV Segment	Functional Class	Viral Protein(s) and Proposed Function(s)
1	Structural protein	Polymerase basic 2 (PB2] protein, together with PB1 and PA, forms the RNA-dependent RNA polymerase (RdRp) complex, which is important for virus replication [51]. Initiates transcription through cap-snatching in conjunction with the proteolytic activity of the PA subunit [52,53,54].
2	Structural protein	Polymerase basic 1 (PB1) protein, the catalytic subunit of the RdRp complex involved in initiation and elongation of mRNA, complimentary (c)RNA, and genomic vRNA [43,55].
2 ^+^	Structural protein	PB1-F2, a viral virulence factor known to promote inflammation and cell death while also enhancing viral RdRp activity [56].
3	Structural protein	Polymerase acidic (PA) protein has proteolytic activity required for the degradation of host‘s RNA polymerase II (RNAP II) during infection [57]. It is also involved in virus assembly [58].
3 ^+^	Structural protein	PA-X contributes to viral growth and repression of host antiviral and protein synthesis [59,60].
4	Surface glycoprotein	Hemagglutinin (HA), an important virus surface glycoprotein that determines host range through binding of virus to susceptible host cells [61].
5	Structural protein	Nucleoprotein (NP) directly binds to newly synthesized viral (v)RNA, providing structural framework for the vRNP complexes [62].
6	Surface glycoprotein	Neuraminidase (NA), another important surface glycoprotein, facilitates virus release and efficient spread of the progeny virus from cell to cell [63,64].
7	Structural protein	Matrix protein 1 (M1) serves as the docking site for vRNPs and nuclear export protein (NEP), playing an important role in shuttling of vRNPs between the nucleus and the cytoplasm. It is also involved in the assembly of vRNPs [65,66].
7 *	Surface glycoprotein	Matrix protein 2 (M2) serves as a membrane ion channel housed within the lipid bilayers of the virus, where it facilitates the acidification process required for the release of the budding virion [67].
8	Nonstructural protein	Nonstructural 1 (NS1) protein plays a critical role in viral replication and pathogenesis. It suppresses splicing of cellular mRNA and prevents nuclear export of polyadenylated cellular RNA [68,69]. It also inhibits host innate immune response, particularly the type I interferon (IFN) system, thereby promoting efficient viral replication and viral cell-to-cell spread [70,71,72].
8 *	Structural protein	NEP, in conjunction with M1, facilitates the nuclear export of vRNP complexes to the cytoplasm of infected cells [43].

EIV segment numbers correspond to the protein structures shown in Figure 3A. * splice variant, ^+^ Frameshift variant.

**Table 3 viruses-13-01657-t003:** Selected examples of commercially available EI vaccine technologies.

Technology	Trade Name	Company	Adjuvant	EIV Strain	Route
Whole inactivated/Subunit ISCOM/ISCOMMatrix	Duvaxyn IE-T^®^	Elanco Animal Health (Buenos Aires, Argentina)	Carbomer, Aluminum Hydroxide	-A/Equi-1/Prague/56 (H7N7) -A/Equi-2/Newmarket/1/93 (H3N8) -A/Equi-2/Suffolk/89 (H3N8) -Anatoxine tétanique	I.M.
	Equip-FT^®^	Pfizer Ltd. (New York, New York, US)	Self-adjuvanting (ISCOM)	-A/Equi-1/Newmarket/77 (H7N7) -A/Equi-2/Borlange/91(H3N8) -A/Equi-2/Kentucky/98 (H3N8) -Anatoxine tétanique	I.M.
	Equilis^®^ Prequenza-TE	Merck Animal Health Inc. (Madison, New Jersey, US)	ISCOMATRIX^TM^	-A/Equi-2/South Africa/4/03 (H3N8) -A/Equi-2/Newmarket/2/93 (H3N8) -Anatoxine tétanique	I.M.
Modified live-attenuated EI vaccine	Flu Avert^®^ I.N.	Merck Animal Health Inc. (Madison, New Jersey, US)	NA	Attenuated, cold-adapted EIV: Kentucky/91 (H3N8)	I.N.
Recombinant virus vector vaccine	Proteqflu-TE^®^	Boehringer Ingelheim (Ingelheim, Germany)	Carbomer	-A/Eq/Ohio/03 (H3N8) -A/Eq/Richmond/1/07 (H3N8) -Anatoxine tétanique	I.M.

NA, not applicable; I.M., intramuscular; I.N., intranasal.

**Table 4 viruses-13-01657-t004:** Advantages and disadvantages of different EI vaccine technologies compared to conventional inactivated-virus vaccines.

EI Vaccine	Advantage	Disadvantage
Subunit vaccines	Nonpathogenic, hence no risk of infection or clinical signs due to vaccination Safer, more stable, and easier to handle compared to MLV The risk of side effects is minimized	Immunity is short-lived and primarily an antibody-based response It requires adjuvant to improve immunogenicity Requires booster doses
Recombinant virus vector vaccines	Long-term immune response Safer, more stable, and easier to handle compared to MLV High immunogenicity No or minimal postvaccination reaction or clinical signs (adverse effects) The immune response can be directed at a specific viral antigen of interest Easy to produce and transport compared to MLV	Requires booster vaccination Could lead to the generation of replication-competent virus Potential of mammalian orthopox to induce tumorigenesis Risk of pathogenicity in some immunocompromised horses Presence of pre-existing immunity to the vector Low titer production
DNA-based vaccines	Nonpathogenic, hence no risk of infection or clinical signs due to vaccination Inexpensive, safer, more stable, and easier to handle compared to MLV Easy to store and transport Similar to recombinant vaccines, the immune response to DNA-based vaccines can be directed at a specific viral antigen of interest Unlike MLV vaccines, DNA-based vaccines are heat stable (temperature-insensitive) Easy to develop and produce using reverse genetics Could be modified as broad-spectrum vaccines by pooling vaccine plasmids together to target multiple EIV antigens	DNA-based vaccines may have relatively poor immunogenicity compared to MLV vaccines Require booster vaccination
Modified live-attenuated vaccines	Induce strong, long-lasting cross-protection. MLV stimulate the immune system better than other EIV vaccines Immunity could last up to 12 months MLV does not need to be potentiated by adjuvants to induce a robust and prolonged protective immunity in horses Elicit local mucosal immunity as well as systemic B- and T-cell proliferation Can be developed via Reverse genetics Can be administered via the intranasal route	Cannot be used in immunocompromised horses Possibility of reversion to virulence Could cause clinical signs or mild infection in some cases when not properly administered

**Table 5 viruses-13-01657-t005:** Scoring system of clinical signs in horses infected with EIV during experimental challenge.

Clinical Sign	Description	Score
Coughing	No cough during daily observation period	0
	Coughing once during daily observation	1
	Coughing twice or more during observation	2
Nasal discharge	No discharge	0
	Serous discharge	1
	Mucopurulent discharge	2
	Profuse mucopurulent discharge	3
Respiration	Normal (<36/min)	0
	Abnormal (dyspnea, tachypnea > 36/min)	1
Depression	No depression	0
	Depression present (lethargy, inappetence)	1

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
