# Peer review of "Equine Influenza Virus and Vaccines"

_viruses, 2021, doi:10.3390/v13081657_

Round 1

Reviewer 1 Report

I have carefully read the manuscript entitled "Equine influenza virus and vaccines" that reviews the evolution of EIV and vaccine strategy. And it contains an in-depth review of the EI virus and vaccine.

However, there are several points to be addressed before its publication.

1.There is so much content tied to the title, “Recent outbreaks of EIV”. So, it seems like a good idea to break it down into several smaller titles.

2. It would be more clear if the author line up "infected animal' and 'source' lines in table 1 and others

3. In the conclusion part, the future strategy part of the vaccine provide limited information. It would be better to suggest in more detail what direction vaccine development is needed.

Author Response

Responses to reviewers’ comments:

Reviewer #1

1)      There is so much content tied to the title, “Recent outbreaks of EIV”. So, it seems like a good idea to break it down into several smaller titles.  To address this, we have inserted some subtitles to break this into sections.

2)      It would be more clear if the author line up "infected animal' and 'source' lines in table 1 and others.  Regarding this and Reviewer #2’s comment, in our copy the table columns are all properly aligned, and we don’t understand the reviewers’ problems.  We wonder whether, in their printing the document, Word has squeezed the table formatting to fit their printer format (e.g. printing a landscape-oriented table in portrait orientation).

3)       In the conclusion part, the future strategy part of the vaccine provide limited information. It would be better to suggest in more detail what direction vaccine development is needed.  To address this, we have altered the Conclusion to bring out these desired aspects of Challenges and Future Directions in, we hope, a more satisfactory manner.

Reviewer 2 Report

Review comments (Oladunni et al, 2021):

1)        Formatting of tables throughout the work were distracting at best.  Please reformat with the goal of (a) eliminating “bullet point” dots found in most tables, (b) spacing between blocks of text within a column when listing topics or descriptions, (c) left-justify text within columns, and (d) avoiding word breaks and hyphenation between lines of texts within a column.

2)        If possible, I would like to see some correlation between the gene segment cartoon in figure 3A and the list of EIV segments in Table 2 in order to clarify both.

3)        The statement in parens, lines 214 and 25, needs to be (a) stated without parens, and (b) supported by a citation.  This statement may change over time, and the current state of affairs should be captured in a time-relative manner.

4)        In table 4, the first disadvantage listed for recombinant virus vector vaccines mentioned is “potential to induce tumorigenesis”.  This statement would refer to adenoviral vectors, but such a vaccine is not described for EIV in this review.  The statement would not apply to the canarypox vector.  Some mammalian Orthopox have been association with epidermal neoplasia in mammals, while to my knowledge none of the Avipox group have been implicated in tumor formation in mammals.  Some recognition of this difference, and the safety of “abortive infection” of Avipox in mammalian systems should be included both in this table and in the text.

5)        The citation listed for canarypox safety (line 401, reference #128) appears to deal with vaccinia vectors in a human vaccine.  Please recheck this reference to confirm that it actually discusses the safety of canarypox expression systems.

6)        The website address on lines 608-609 seems awkward within the text, and may be better placed in the references.

7)        In table 5, you refer to both “dyspnea” and “tachypnoea”.  One is American English, one is British English.  Either word ending is correct, but the same form should be used for both terms to remain consistent.

8)        The multiple online references stated in lines 766 through 769 should be listed in their complete web address URL form, and ideally presented in the reference section.

9)        The online resource stated on lines 846-847 is presented as a live link, and not a web address URL.  Please move to the complete URL citation to the reference section, as with other online references.

Author Response

Reviewer #2:

1)        Formatting of tables throughout the work were distracting at best.  Please reformat with the goal of (a) eliminating “bullet point” dots found in most tables, (b) spacing between blocks of text within a column when listing topics or descriptions, (c) left-justify text within columns, and (d) avoiding word breaks and hyphenation between lines of texts within a column.  To address this, we have (a) removed the bullet-point dots and (b) added spacing lines.  Regarding (c) and (d), as mentioned above (Reviewer #1’s comment #2), we do not see the problems the reviewers see. Some of the tables are in landscape orientation and the problems might be due to Word’s printing these in portrait orientation.

2)        If possible, I would like to see some correlation between the gene segment cartoon in figure 3A and the list of EIV segments in Table 2 in order to clarify both.  To address this, we have modified Figure 3A by adding the gene segment numbers, and noted the correlation in the Legends to Figure 3 and Table 2.

3)        The statement in parens, lines 214 and 25, needs to be (a) stated without parens, and (b) supported by a citation.  This statement may change over time, and the current state of affairs should be captured in a time-relative manner.  The reviewer is correct, this situation could change at any time, and to remedy this we have deleted the parenthetical phrase entirely (Line 214). 

4)        In table 4, the first disadvantage listed for recombinant virus vector vaccines mentioned is “potential to induce tumorigenesis”.  This statement would refer to adenoviral vectors, but such a vaccine is not described for EIV in this review.  The statement would not apply to the canarypox vector.  Some mammalian Orthopox have been association with epidermal neoplasia in mammals, while to my knowledge none of the Avipox group have been implicated in tumor formation in mammals.  Some recognition of this difference, and the safety of “abortive infection” of Avipox in mammalian systems should be included both in this table and in the text.  The reviewer is correct, and we have modified Table 4 to reflect that the disadvantage specifically applies to mammalian orthopox vectors. We also modified the text at Line 401 to mention the avipox abortive infection.

5)        The citation listed for canarypox safety (line 401, reference #128) appears to deal with vaccinia vectors in a human vaccine.  Please recheck this reference to confirm that it actually discusses the safety of canarypox expression systems.  The reviewer is correct and we have modified the line (398) that cites reference 128 to make it more accurately describe vaccinia vectors. We also added a new reference 129 to support the safety of the canarypox vector.

6)        The website address on lines 608-609 seems awkward within the text, and may be better placed in the references.  We have moved it into the references (#193)

7)        In table 5, you refer to both “dyspnea” and “tachypnoea”.  One is American English, one is British English.  Either word ending is correct, but the same form should be used for both terms to remain consistent.  We have changed tachypnea into American usage for consistency.

8)        The multiple online references stated in lines 766 through 769 should be listed in their complete web address URL form, and ideally presented in the reference section.  We have moved these into the references (#221, 222).

9)        The online resource stated on lines 846-847 is presented as a live link, and not a web address URL.  Please move to the complete URL citation to the reference section, as with other online references.  We have moved this into the references (#236).